# Learning to Drive Anywhere

**Ruizhao Zhu**    **Peng Huang**    **Eshed Ohn-Bar**    **Venkatesh Saligrama**
Boston University

**Abstract:** Human drivers can seamlessly adapt their driving decisions across geographical locations with diverse conditions and rules of the road, e.g., left vs. right-hand traffic. In contrast, existing models for autonomous driving have been thus far only deployed within restricted operational domains, i.e., without accounting for varying driving behaviors across locations or model scalability. In this work, we propose AnyD, a single geographically-aware conditional imitation learning (CIL) model that can efficiently learn from heterogeneous and globally distributed data with dynamic environmental, traffic, and social characteristics. Our key insight is to introduce a high-capacity geo-location-based channel attention mechanism that effectively adapts to local nuances while also flexibly modeling similarities among regions in a data-driven manner. By optimizing a contrastive imitation objective, our proposed approach can efficiently scale across the inherently imbalanced data distributions and location-dependent events. We demonstrate the benefits of our AnyD agent across multiple datasets, cities, and scalable deployment paradigms, i.e., centralized, semi-supervised, and distributed agent training. Specifically, AnyD outperforms CIL baselines by over 14% in open-loop evaluation and 30% in closed-loop testing on CARLA.

**Keywords:** Global-scale Autonomous Driving, Imitation Learning, Transformer

## 1  Introduction

Driving at scale involves a complex and nuanced decision-making process across diverse social and environmental conditions. For instance, a driving model for turning at intersections in San Francisco, CA may be able to generalize relatively well when deployed in Washington, DC, over 2400 miles away. However, when deployed just north of it in New York City, the model would discover that right turns on red, which have been banned in the city [1], could result in social disruption and potentially unsafe conditions at intersections. When deployed at intersections in Pittsburgh, PA, the model may begin accelerating at a green light only to be confounded by the frequent occurrence of the Pittsburgh left [2], resulting in frequent and uncomfortable braking. These examples illustrate how the lack of modeling of location-based traffic behavior and social norms can lead to potentially safety-critical consequences. Beyond city-level variability, failing to accurately account for state or country-level differences in traffic regulations and social norms can also have dire consequences, e.g., from the directionality of travel [3] to varying maximal speed limitations [4] or yielding expectations [5]. How can we design and learn models that flexibly accommodate the heterogeneous data encountered across challenging and diverse geographical, environmental, and social conditions?

Despite recent advances in decision-making models for autonomous driving, models are often trained and evaluated within limited operational domains, i.e., a handful of geographical regions and social conditions (e.g., Waymo's service in Pheonix, AZ, and San Francisco, CA [6]). Autonomous driving benchmarks are often collected in a handful of cities and routes [7–10]. Existing frameworks for learning to drive (e.g., [11–15]) train a single policy without considering geo-location or policy adaptation. While such methods may be used to train and adapt different regional models, there are often many similarities which can be shared among the different locations and benefit potentially small and rare datasets with imbalanced distributions. In this work, we propose an approach for

7th Conference on Robot Learning (CoRL 2023), Atlanta, USA.

learning to modulate predictions across settings and locations, i.e., *even in seemingly similar visual settings*, within a single driving agent.

While prior works have leveraged transfer learning [16–23], e.g., through access to unlabeled data of a target domain, or introduced internal layers that learn to adapt model output across various domains [24–28, 16, 29–31], these methods have exclusively focused on low-level object classification and detection tasks, and have not explicitly accounted for geographical priors or reasoning. In contrast, we study end-to-end models for learning safe perception and decision-making in intricate 3D navigation scenarios. In this case, to avoid a potential accident, perception and action characteristics must both be carefully tuned to consider geographical location when reasoning over traffic maneuvers and predicting social behavior. Moreover, the training process of our sensorimotor models may require order-of-magnitude higher sample complexity, i.e., due to the higher rarity of policy-level events and intricate maneuvers [32]. Thus, geo-aware model capacity should be explored jointly with approaches for efficient adaptation and parameter sharing, as we do in this work.

**Contributions:** We make **three key contributions** towards autonomous systems at scale: 1) We revisit current end-to-end driving models to identify limitations in learning from heterogeneous and distributed data sources. In particular, we build on recent advances in transformer-based models [33, 34] for learning high-capacity, geo-aware imitation learning agents that can adapt across geographical locations while sharing parameters and computation within a single network. 2) To facilitate efficient training across inherently imbalanced data distributions and maneuvers, we further generalize conditional imitation learning by designing a *supervised contrastive loss* over conditional commands and locations. 3) We combine three public autonomous driving datasets collected by different companies and platforms across 11 locations to extensively evaluate the impact of the proposed scalable learning framework. To understand generalization across diverse use-cases and model training regimens, we comprehensively analyze the benefits of our framework for various scalable deployment scenarios, including centralized (i.e., within a single company or server with shared raw data logs), distributed (i.e., with scalable federated computation), and semi-supervised (i.e., with unlabeled data) training.

## 2   Related Work

**Learning to Drive from Demonstrations:** Despite impressive recent advances in learning to drive, approaches often leverage simple navigation tasks, i.e., lane following, intersection turning, and basic collision avoidance (e.g., with CIL [12, 35, 15, 36–39, 13]), or short real-world routes in a handful of locations (e.g., [40, 11, 41, 42, 14, 43, 10, 44–46, 7]). We note that GPS localization in prior approaches may only be used to determine a next *high-level command at an intersection* [35], and not to learn regionally or socially appropriate decisions. Yet, training models among locations without such geo-awareness results in an ill-posed problem with ambiguous samples. Thus, our work can be seen as a natural generalization of goal-conditional imitation learning frameworks [35, 47, 13] to incorporate geographical information for learning a high-capacity and controllable model.

**Domain Adaptation:** Model adaptation, i.e., from a source to a target domain with unlabeled data, has mostly focused on segmentation and detection tasks [16–22]. However, the robustness and reliability of current domain adaptation techniques at large scale have been repetitively questioned [48–50]. Moreover, the aforementioned techniques have not been previously studied within the more complex end-to-end training paradigm for decision-making models. Particularly relevant to our study are approaches that learn universal object detection models [24, 26] via self-attention and weighing feature channels based on the output of multiple parallel layers (i.e., adapters [51]). In contrast, our proposed *cross-attention-based* network architecture can more effectively fuse geographically-derived and visual features while also outperforming adapter-based methods [24].

**Benefits of Contrastive Learning:** Researchers have been increasingly exploring the benefits of contrastive learning frameworks for learning generalized representations, even under imbalanced

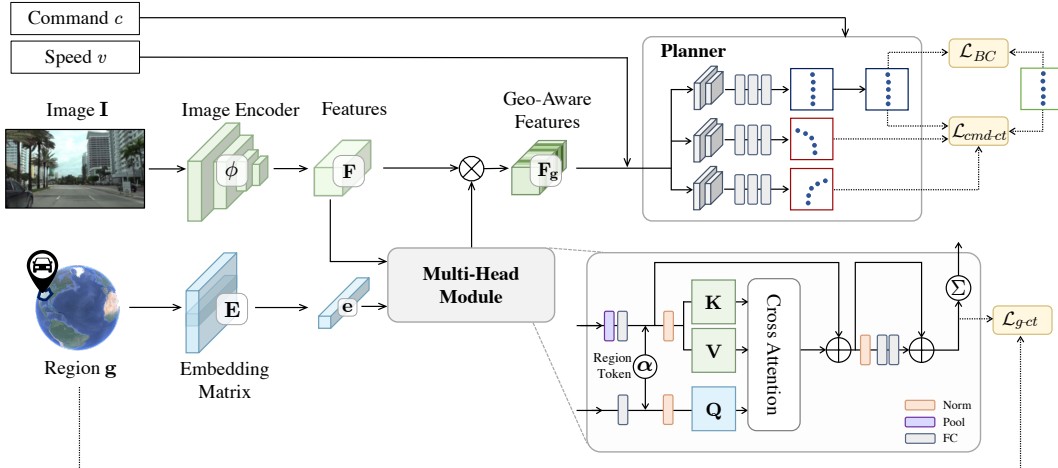

Figure 1: **Model Overview.** Our model maps image, region, speed, and conditional command observations to future decisions, parameterized as waypoints in the map view. To efficiently learn a high-capacity model, we leverage a multi-head cross attention module which fuses and adapts internal representations in a geo-aware manner. Our imitation objective, defined over human-demonstrated waypoints (outlined in green), the other command branches (outlined in red), and the predicted weights by the multi-head module, regularizes model optimization under diverse data distributions.

or long-tail settings [52–56]. Our main use-case inherently involves learning over diverse and imbalanced underlying data distributions. For instance, a Tesla may suddenly trigger a warning in a challenging scenario or an unsupported region, in which case small amounts of demonstration data from the driver may be collected and available for training. Moreover, although diverse and rare traffic scenarios can occur within any local city region or country, the *underlying distribution of such events* can significantly shift among locations. Recently, Mandi et al. demonstrated the benefits of *unsupervised contrastive learning* for improved imitation learning within simple robotic use-cases [57]. Instead, we demonstrate the benefits of *supervised contrastive learning* techniques (e.g., [55]) by designing a novel loss function for conditional imitation learning frameworks at scale.

**Distributed Learning to Drive:** We comprehensively analyze our proposed approach across training paradigms suitable for scalable deployment in order to ensure the generalization of our findings. In particular, Federated Learning (FL) provides a natural framework for implementing AnyD in the real-world. The goal of FL to drive is to train a global model leveraging distributed data and models from different agents [58], i.e., where agents may avoid sharing raw driving logs and data due to various privacy and efficiency considerations. However, dealing with data heterogeneity among agents [59–62] remains a challenge. We demonstrate our novel geo-conditional mechanism to complement current federated learning algorithms. Somewhat surprisingly, our FL model variants result in outperformance compared to the centralized-trained counterparts due to the effective regional bias handling. We note that this is without having to share potentially sensitive geographical information, as our embedding matrix (defined in Sec. 3) is kept local and private in our implementation.

## 3   Method

We propose a geo-conditional agent (AnyD) which generalizes existing conditional imitation learning methods [35, 63] through two key aspects. First, we propose a novel network structure that leverages a *multi-head transformer module* for geo-aware adaptation of visual features across regions (Sec. 3.2). Second, we design a *contrastive learning objective* which regularizes training and addresses imbalances across locations and capture settings (Sec. 3.3). An overview of our approach is depicted in Fig. 1.

### 3.1 Problem Definition

Our objective is to learn a goal-directed agent that can effectively reason over varying traffic rules and social norms in complex and dynamic real-world settings. We leverage offline approaches relying on learning from driver demonstrations [64, 65, 40, 66, 67] as they can safely learn to map sensor observations to actions, i.e., as opposed to interactive methods [68, 69, 36, 70, 71]. As an example use-case, consider a deployed Tesla or Waymo fleet encountering challenging settings beyond its current constrained and geo-fenced deployment [72, 73]. Here, a human can take-over and demonstrate desired driving behavior which can subsequently be uploaded to a shared cloud server (i.e., centralized training) or updated to improve the model locally (i.e., federated training, we consider both cases in Sec. 3.4). However, current end-to-end agents that learn to drive in a data-driven manner, e.g., based on CIL [35, 12, 74, 13], do not differentiate among regional norms.

**Geo-Conditional Imitation Learning:** We assume a dataset of demonstrations $\mathcal{D} = \{(\mathbf{x}, \mathbf{y})\}_{i=1}^{N}$, i.e., measurements of $\mathbf{x} = (\mathbf{I}, c, v, \mathbf{g}) \in \mathcal{X}$, where $\mathbf{I} \in \mathbb{R}^{W \times H \times 3}$ is an image of the current environment, $v \in \mathbb{R}$ is the speed, $c \in \mathbb{N}$ is a navigation command [35, 12], $\mathbf{g} \in \{0, 1\}^{G}$ is a region index encoded as a one-hot vector over a total $G$ regions, and corresponding action labels $\mathbf{y} \in \mathcal{Y}$ based on human drivers. Consistently with prior work [75, 37, 76], we predict a waypoint-based label in the bird's eye view over the next five planned locations (2.5 seconds), such that $\mathbf{y} = \{\mathbf{w}_t\}_{t=1}^{5}$ and $\mathbf{w}_i \in \mathcal{R}^2$. The high-level waypoint output in the bird's eye view can also help standardize policy decisions across globally distributed platforms with heterogeneous sensor configurations [41]. In our work, we experiment with various definitions for $\mathbf{g}$ (manually defined city labels and unsupervised neighborhood-level labels, these do not require precise GPS localization). Moreover, we note AnyD does not rely on image-level perception labels or high-definition map information. As such, it can be trained based on cheaply collected GPS-based waypoint labels and benefit from rapid advancements in positioning technology (analysis of localization noise when training AnyD models can be found in the supplementary). We train a geographically-aware policy function $\pi : \mathcal{X} \to \mathcal{Y}$ using supervised learning [12, 13]. Learning the policy $\pi$ in geo-conditional imitation learning requires carefully fusing image and geo-location information, i.e., as opposed to just basic concatenation. Next, we introduce our network architecture which efficiently generalizes the branch-based architectures of CIL-based approaches [35].

### 3.2 Geo-Conditional Transformer Module

To learn a scalable policy function, i.e., across cities, countries, and platforms, we design a single network which adapts its decisions based on an efficient multi-head module. The mechanism is motivated by transformer [34, 33, 77], with three main aspects. First, the queries are only conditioned on the part of the input, i.e., the region-based features. Second, we do not use spatial attention as in ViT-type architectures [33], but instead learn a low-dimensional channel weight vector which can be trained efficiently [77]. Third, while multi-head mechanisms have been used by prior methods [33], we propose to jointly predict a scalar weight for each head prior to the summation of the heads. This formulation is analogous to a mixture or adapter-based model [78, 28]. Our domain attention mechanism is implemented via a *region token* that enables the model to specialize the heads to specific domains or tasks and subsequently combine the heads based on the current appropriate region and decision. For instance, we identify the emergence of traffic rules, such as left vs. right-hand driving when inspecting the output of the learned heads in Sec. 4.

Our model first extracts image features from the input image $\mathbf{I}$. Subsequently, a multi-head transformer module computes channel weights for the features based on the current region definition $\mathbf{g}$. Finally, the planner utilizes the re-weighted geo-aware features $\mathbf{F_g}$ and speed information $v$ to generate waypoints for different commands. The command input $c$ then selects the required waypoints $\hat{\mathbf{y}}$ for execution. The multi-head transformer module takes as input *visual features* extracted using a ResNet-34 encoder $\phi$ [79], $\mathbf{F} = \phi(\mathbf{I}) \in \mathbb{R}^{8 \times 13 \times C}$ and a *regional embedding* $\mathbf{e} = \mathbf{g}^{\top} \mathbf{E}$ (assuming a column vector $\mathbf{g}$), extracted from a trainable embedding matrix $\mathbf{E} \in \mathbb{R}^{G \times C}$ [80]. We use $C = 512$ such that the output of the multi-head module is a 512-dimensional vector for weight-

ing each channel in $\mathbf{F}$ and computing the geo-aware features $\mathbf{F_g}$ using the weighted and summed $H$ output heads (Eqn. 3). The visual features are pooled (to accommodate the channel-wise attention), processed through a Fully Connected (FC) layer, and concatenated with a *region token* $\boldsymbol{\alpha}$, $\mathbf{z_I} = [\boldsymbol{\alpha}, \text{FC}((\text{Pool}(\mathbf{F}))] \in \mathbb{R}^{(C+1)\times d}$, where $d = 128$ sets the number of hidden units. $\boldsymbol{\alpha}$ will be updated and used to weigh the multiple heads at the output of the module, as shown in Eqn. 3. Similarly, the region embedding $\mathbf{z_g} = [\boldsymbol{\alpha}, \text{FC}(\mathbf{e})] \in \mathbb{R}^{(C+1)\times d}$. The computation steps for the geo-aware transformer can then be summarized as:

$$\mathbf{z} = \mathbf{z_I} + \text{Attention}(\text{LN}(\mathbf{z_I}), \text{LN}(\mathbf{z_g})) \tag{1}$$

$$\hat{\mathbf{z}} = \mathbf{z} + \text{MLP}(\text{LN}(\mathbf{z})) \tag{2}$$

$$\mathbf{F_g} = \sum_{h=1}^{H} (\hat{\boldsymbol{\alpha}}_h \hat{\mathbf{z}}_{h,2:C+1}) \otimes \mathbf{F} \tag{3}$$

where LN denotes Layer Normalization, $\otimes$ denotes channel-wise multiplication, and $\hat{\boldsymbol{\alpha}}$ is the updated region token values. $\mathbf{z}$ in the second step is pooled before addition to make the shape consistent. We follow ViT [33] to compute attention as

$$\text{Attention}(\mathbf{z_I}, \mathbf{z_g}) = \text{softmax}\left(\frac{\mathbf{QK}^T}{\sqrt{d}}\right)\mathbf{V} \tag{4}$$

where $\mathbf{Q} = \mathbf{z_g}\mathbf{W}^Q$, $\mathbf{K} = \mathbf{z_I}\mathbf{W}^K$, $\mathbf{V} = \mathbf{z_I}\mathbf{W}^V$ and $\mathbf{W}^Q \in \mathbb{R}^{d\times d}$, $\mathbf{W}^K \in \mathbb{R}^{d\times d}$ and $\mathbf{W}^V \in \mathbb{R}^{d\times d}$ are learned matrices. Unlike ViT, we do not merge the multiple heads by concatenating such that there are $H$ outputs, each $C+1$-dimensional, i.e., $\hat{\mathbf{z}} \in \mathbb{R}^{H\times(C+1)}$. Here, the weights for the $h$-th head are stored at the first index of the head output vector, i.e., $\hat{\boldsymbol{\alpha}}_h = \hat{\mathbf{z}}_{h,1}$. The adapted geo-aware features are then given to a command-conditional branch as shown in Fig. 1 for predicting the final waypoints. To optimize the network, we leverage a contrastive loss function over maneuvers and regional decisions, as discussed next.

### 3.3 Contrastive Imitation Learning

**Loss Function:** Standard supervised learning approaches for imitation learning leverage an $L_1$ loss, i.e., behavior cloning, between predicted and demonstrated ground-truth waypoints [71, 12]. However, in our case of highly heterogeneous and imbalanced data over maneuvers and regions, this loss can result in poor performance and overfitting to local biases [81, 55]. In particular, the distribution of both conditional commands $c$ and regions $\mathbf{g}$ can be highly skewed, with certain critical events (e.g., turns) occurring at a much lower frequency. While we employ a branched architecture [35, 13], a subset of the branches may be trained over a fraction of the total samples, i.e., with most updating the 'forward' branch. We hypothesize that such imbalances can introduce noisy predictions from poorly-trained branches. When adding the additional complexity of learning region-conditional policies, issues in data imbalance and heterogeneity compound. To tackle this practical safety-critical issue, Chen et al. [65] employed a privileged teacher (i.e., learned from complete ground truth observations of the 3D surroundings instead of raw images) that can be used for additional sampling and data augmentation. However, training such a privileged expert requires extensive annotation of real-world data, which is not scalable. Instead, we propose to introduce command and region-contrastive objectives as a simple and effective strategy for improving model optimization and providing more supervision when handling imbalanced data. In our analysis, we demonstrate the utility of this approach for both vanilla CIL and the proposed geo-CIL. As far as we are aware, we are the first to empirically analyze such benefits for imitation-learned driving agents at scale.

We propose to incorporate two additional terms in addition to the main behavior cloning loss, $\mathcal{L}_{BC}$. The total loss can be computed as

$$\mathcal{L} = \mathcal{L}_{BC} + \lambda_c \mathcal{L}_{cmd-ct} + \lambda_g \mathcal{L}_{g-ct} \tag{5}$$

where $\lambda_c$, $\lambda_g$ are hyperparameters, $\mathcal{L}_{cmd-ct}$ and $\mathcal{L}_{g-ct}$ are contrastive losses. Next, we define each of the proposed loss terms.

**Command Contrastive Loss:** The branched conditional command architecture of CIL updates each branch based on a subset of the data samples in each batch $\mathcal{B}$. As some commands are highly underrepresented in natural driving data, we propose a command contrastive loss that leverages *predictions for other commands for the same sample* as negative examples,

$$\mathcal{L}_{cmd-ct} = -\frac{1}{|\mathcal{B}|} \sum_{i \in \mathcal{B}} \log \frac{\exp(d(\hat{\mathbf{y}}_i^{c_i}, \mathbf{y}_i)/\tau)}{\sum_c \exp(d(\hat{\mathbf{y}}_i^c, \mathbf{y}_i)/\tau)} \tag{6}$$

where the loss is computed over a single positive example with prediction outputted by the ground truth command branch $c_i$, and the other predictions (i.e., hypothetical commands) as negatives. $d$ is a similarity function (we use negative $L_2$ distance) and $\tau \in \mathbb{R}^+$ is a scalar temperature parameter. The command contrastive loss is a natural extension supervised contrastive learning [55] to our case of conditional imitation learning, with two key differences. First, we do not apply it to the feature space (as commonly done) but instead to the *output space* of the network, i.e., in order to better model differences among maneuvers. Second, the command contrastive loss is not computed over all other samples in the same batch with different commands as in standard supervised contrastive loss [55]. While this practice may work for simple classification tasks, in our case such other samples tend to also involve different driving situations. We found leveraging other samples in this manner when training an imitation model to degrade policy performance, most likely due to the added learning complexity and ambiguity. A similar reasoning can be applied to improve optimization of the geo-conditioned transformer module, as discussed next.

**Geo-Contrastive Loss:** While driving behavior across cities and regions can often be similar, in practice, the cities in our employed datasets (detailed in Sec. 4) all have unique local characteristics effectively modeled. Thus, we also propose to incorporate a region (i.e., city)-based contrastive loss. We apply the loss within the transformer module over different output head weights $\hat{\boldsymbol{\alpha}} \in \mathbb{R}^H$. Here, we follow standard contrastive loss implementation [55] and select the $i$-th sample as an anchor. During training, for the $i$-th sample in a batch, positive samples $\mathcal{P}(i)$ are defined within the same city, while negative samples $\mathcal{N}(i)$ from differing cities,

$$\mathcal{L}_{g-ct} = -\frac{1}{|\mathcal{B}|} \sum_{i \in \mathcal{B}} \frac{1}{|\mathcal{P}(i)|} \log \frac{\sum_{p \in \mathcal{P}(i)} \exp(d(\hat{\boldsymbol{\alpha}}_i, \hat{\boldsymbol{\alpha}}_p)/\tau)}{\sum_{a \in \mathcal{A}(i)} \exp(d(\hat{\boldsymbol{\alpha}}_i, \hat{\boldsymbol{\alpha}}_a)/\tau)} \tag{7}$$

where $\mathcal{A}(i) \equiv \mathcal{P}(i) \cup \mathcal{N}(i)$ and $d$ is a similarity function (we use negative $L_2$ distance).

### 3.4 Scalable Training Settings

We comprehensively analyze the training of our model using three different scalable deployment settings. First, in ***Centralized Learning (CL)***, agents are able to share all sensor data and geographical information with a centralized server. Consequently, the server conducts supervised learning of our AnyD model over all of the raw data. To further analyze model scalability, we implement a ***Semi-Supervised Learning (SSL)*** model, which can leverage ample unlabeled data that may be available across locations (we follow [13]). Finally, we study the applicability of our findings within federated learning approaches, as sharing raw sensor data can be inefficient or even potentially undesirable. For instance, our AnyD agent may be distributed over numerous heterogeneous data sources with various constraints, i.e., local regulations, legal authorities, and privacy requirements or preferences. Thus, we also analyze a ***Federated Learning (FL)*** agent which does not require sharing raw and geographical information with a server. To fully understand the role of our proposed network structure within such paradigms, we optimize AnyD using two federated learning algorithms, FedAvg[58] and FedDyn[59]. We note that the geographical embedding matrix $\mathbf{E}$ which contains city-level information remains locally updated on each agent (i.e., akin to a form of local model personalization). In this manner, $\mathbf{E}$ reduces to a row vector as the embedding vector for the specific region (city in our implementation). We note that this results in the removal of the geo-contrastive loss term $\mathcal{L}_{g-ct}$ in Eqn. 7. The supplementary contains additional details regarding our implementation.

# 4 Experiments

In this section, we first introduce our combined multi-city benchmark extracted from multiple publicly available datasets. Specifically, we present ablation studies for various model design choices and loss terms. To understand the benefits of the proposed framework on various training schemes, we also report analysis with three different training paradigms. This ensures our model and findings are relevant across real-world use-cases, e.g., with efficient distributed settings at large-scale.

## 4.1 Datasets and Metrics

To learn a global scale driving policy, we conduct training on data from three different datasets including Argoverse 2 (*AV2*) [9], nuScenes (*nS*) [10] and Waymo (*Waymo*) [7]. While these datasets do not have any official waypoints prediction benchmark, we extract these from the provided raw data logs. Specifically for each frame, we processing the raw data to get the future 2.5s as ground truth waypoints, current velocity, a front-view RGB image, navigational commands and a city-level information. The data spans 11 cities. We split the data into training, validation and testing data. Our split utilizes 190k, 20k, and 35k training samples for AV2, nS and Waymo datasets respectively. We follow standard evaluation using Average $L_2$ Displacement Error (ADE) and Final $L_2$ Displacement Error (FDE) over future waypoints in the BEV space. We also evaluate closed-loop policy performance using CARLA [82]. As CARLA benchmarks do not generally involve regional modeling, we introduce left-hand driving and town-varying behavior of agents. Our supplementary provides additional details and experiments.

## 4.2 Results

**Model and Loss Ablation:** We study the underlying architecture of the model in Table 1. We find that replacing intermediate image-level heatmaps (used in several CIL-based baselines [65, 13]) with fully-connected layers provides improved reasoning for our diverse perspective settings (reducing ADE from 1.24 to 1.16). We also demonstrate our geo-conditional transformer framework with three heads to outperform other supervision choices, e.g., embedding concatenation and supervision as an auxiliary prediction task as in Ayush et al. [83] (1.09 vs. 1.15 ADE).

Table 1: **Ablative Studies on Model Architecture, Geo-Conditional Module and Loss.** We start from CIL architectures [65, 13] and gradually add different components losses to get AnyD.

| Ablation | Method | ADE | FDE |
|---|---|---|---|
| CIL Architecture | CIL [65] | 1.32 | 2.55 |
| | BEV Planner [13] | 1.24 | 2.45 |
| | Our Planner | **1.16** | **2.17** |
| Geo-CIL Architecture | Concatenation | 1.14 | 2.12 |
| | Task Supervision [83] | 1.15 | 2.24 |
| | Hybrid ViT [33] | 1.20 | 2.30 |
| | Universal Adapter [84] | 1.10 | 2.11 |
| | Geo Transformer w/ $\mathcal{L}_{BC}$ | **1.09** | **2.08** |
| Loss Function | $\mathcal{L}_{BC}, \mathcal{L}_{cmd-ct}$ | 1.07 | 1.97 |
| | $\mathcal{L}_{BC}, \mathcal{L}_{g-ct}$ | 1.06 | 2.00 |
| | AnyD ($\mathcal{L}$) | **1.05** | **1.93** |

AnyD also outperforms another attention-based method e.g., Hybrid ViT [33] on concatenated image-city features (1.09 vs. 1.20 ADE), which validates its efficiency on adaptation. Moreover, the proposed geo-conditional module can be used to increase the modeling capacity of the agent, and thus can scale beyond simple task supervision. We also find a holistic effect among the proposed loss terms, with a combination leading to the best results (1.93 vs 2.08 FDE for the vanilla behavior cloning loss). Fig. 2 depicts a case where AnyD handles diverse traffic regulations and social norms such as turning right (wider turn) in Singapore and yielding a 'Pittsburgh left.'

**Training Paradigms:** Table 2 reports the impact of various model training schemes on ADE performance (additional details, including FDE-based analysis, are in the supplementary). We observe consistent improvements across paradigms and cities even *with severe data imbalance*. Moreover, leveraging unlabeled YouTube data for each city results in further gains, specifically for cities with lesser data (MTV, PAO, SGP, and PHX). For instance, MTV improves from 1.40 to 1.23 ADE due to the unlabeled data, showing the importance of this mechanism for our use-case. Overall, the model is shown to outperform the baseline of Zhang et al. [13], which does not leverage geo-location. Consistently with prior work, federated learning algorithms tend to under-perform their centralized

Table 2: **Evaluating AnyD with Different Training Paradigms.** AnyD efficiently integrates into various training paradigms (CL-Centralized Learning, SSL-Semi-Supervised Learning, and FL-Federated Learning). ADE is computed across the 11 cities in our dataset. Our planner is our proposed architecture for direct image-to-BEV prediction (without the geolocation information or introduced auxiliary loss terms, see supplementary for additional architecture details).

| Settings | Method | Avg | PIT | WDC | MIA | ATX | PAO | DTW | BOS | SGP | PHX | SFO | MTV |
|---|---|---|---|---|---|---|---|---|---|---|---|---|---|
| CL | CIL [65] | 1.32 | 1.14 | 1.40 | 1.47 | 1.23 | 1.49 | 1.01 | 0.89 | 1.09 | 1.65 | 1.67 | 1.48 |
| | CILRS [12] | 1.27 | 1.18 | 1.28 | 1.43 | 1.16 | 1.52 | 1.11 | 0.84 | 1.02 | 1.39 | 1.60 | 1.40 |
| | BEV Planner [13] | 1.24 | 1.18 | 1.01 | 1.34 | 1.23 | 1.55 | 1.03 | 0.90 | 1.07 | 1.38 | 1.58 | **1.39** |
| | TCP [38] | 1.22 | 1.09 | 1.23 | 1.41 | 1.14 | 1.47 | 0.99 | 0.87 | 1.01 | 1.39 | 1.49 | 1.40 |
| | Our Planner | 1.16 | 1.24 | 1.12 | 1.12 | 1.38 | 1.39 | 1.02 | 0.92 | 1.10 | 1.08 | 0.89 | 1.41 |
| | AnyD | **1.05** | **1.12** | **0.96** | **0.95** | **1.16** | **1.31** | **0.89** | **0.82** | **1.03** | **0.98** | **0.83** | 1.40 |
| SSL | SelfD [13] | 1.02 | 1.13 | 1.03 | 1.01 | 1.25 | 1.26 | 0.93 | 0.80 | 0.95 | **0.82** | 0.79 | 1.29 |
| | AnyD | **0.97** | **1.06** | **0.93** | **0.92** | **1.19** | **1.24** | **0.89** | **0.76** | **0.94** | 0.84 | **0.75** | **1.23** |
| FL | FedAvg [58] | 1.42 | 1.38 | 1.43 | 1.41 | 1.73 | 1.63 | 1.23 | 0.93 | 1.21 | 1.53 | 1.42 | 1.64 |
| | FedDyn [59] | 1.19 | 1.23 | 1.15 | 1.21 | 1.59 | **1.51** | 1.11 | 0.797 | 0.95 | 0.97 | 0.99 | 1.30 |
| | AnyD (FedAvg) | 1.20 | 1.23 | 1.06 | 1.04 | 1.62 | 1.61 | 1.00 | 0.81 | 1.07 | 1.33 | 1.01 | 1.41 |
| | AnyD (FedDyn) | **0.98** | **1.08** | **0.91** | **0.91** | **1.50** | 1.54 | **0.90** | **0.68** | **0.82** | **0.62** | **0.70** | **1.12** |

Table 3: **Closed-Loop Evaluation in CARLA.** We report closed-loop metrics of Success Rate (SR), Route Completion (RC), Infraction Score (IS) and Driving Score (DS) compared to a baseline planner which is not trained in a geo-aware manner.

| Metrics | ADE ↓ | FDE ↓ | SR ↑ | RC ↑ | IS ↑ | DS ↑ |
|---|---|---|---|---|---|---|
| BEV Planner [13] | 0.58 | 0.97 | 0.29 | 0.50 | 0.61 | 0.36 |
| AnyD | **0.46** | **0.79** | **0.36** | **0.69** | **0.67** | **0.50** |

counterparts. Yet, due to better handling of local biases, AnyD is shown to benefit federated learning, surpassing centralized training (0.98 vs. 1.05 ADE).

**Closed-Loop Evaluation:** The results of the closed-loop experiments on CARLA are shown in Table 3. We find consistent improvements in success rate, route completion, and infractions (the supplementary video shows qualitative examples where the baseline model struggles with left-hand driving). We compute both open and closed-loop metrics by saving the expert actions for the test sequences. AnyD outperforms the baseline planner [13] on both open-loop metrics (reducing ADE from 0.58 to 0.46) and closed-loop metrics (improving driving score by 38%, from 0.36 to 0.50). Nonetheless, overall success rates are quite low for our benchmark, as it contains significant behavior variability. This highlights the challenging nature of the region-aware decision-making task for current imitation learning models, either in the real-world or in simulation.

## 5 Conclusion

We envision large-scale navigation agents that can seamlessly operate in heterogeneous and distributed locations. Our work introduces an efficient framework for training a universal high-capacity navigation agent across diverse locations. Using our proposed agent, fleets of vehicles can increasingly grow their operation capacity to novel conditions, i.e., by involving humans and collecting both unlabeled or labeled demonstration data for policy training. Nonetheless, effectively incorporating geo-awareness into driving models remains a challenging and under-explored research problem.

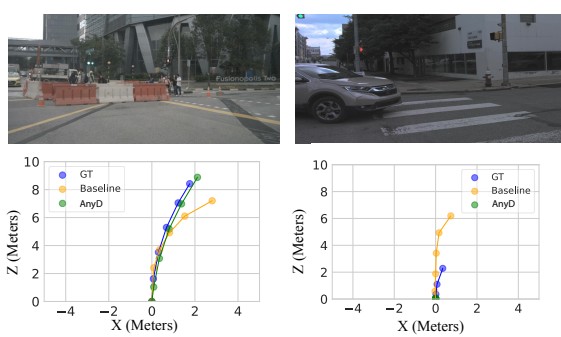

Figure 2: **Qualitative Results.** We plot predicted waypoints in the BEV for comparison. AnyD exhibits robustness in region-specific cases, including turning right (wider turn) in Singapore, yielding a 'Pittsburgh left' vehicle in Pittsburgh.

**Acknowledgments**

This research was supported by a Red Hat Research Grant, Army Research Office Grant W911NF2110246, National Science Foundation grants (CCF-2007350, CCF-1955981, and IIS-2152077), and AFRL Contract no. FA8650-22-C-1039.

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

# 6 Supplementary

We provide additional implementation details, including network architecture and training protocol, as well as additional analysis, including ablative studies, results on CARLA, and additional qualitative examples.

## 6.1 Implementation Details

This section first provides details regarding our proposed network architecture and discuss differences with baseline models (Sec. 6.1.1). Next, we provide details regarding the processing of the driving datasets to construct the multi-city benchmark used throughout the analysis (Sec. 6.1.2). Finally, we discuss evaluation settings (Sec. 6.1.3) and training protocol (Sec. 6.1.4).

### 6.1.1 Architecture and Baselines

We leverage an ImageNet-pretrained ResNet-34 [79] as our backbone $\phi$. All images are resized to $400 \times 225$ prior to being inputted to the model. To leverage diverse camera viewpoints, we build on prior models in conditional imitation learning [65, 13] and train a direct image-to-BEV prediction model, i.e., without assuming a fixed known BEV perspective transform. Moreover, we also find the removal intermediate image-level heatmaps [65, 13] (and directly regressing the BEV waypoints) to improve model performance. Fig. 3 compares our proposed network architecture for image-to-BEV planning to a standard baseline architecture (e.g., [13, 65]). Prior image-based models may utilize deconvolutional layers to obtain an image-aligned heatmap and followed by a soft-argmax ('SA' in Fig. 3) and 2D waypoint projection to the BEV space. The projection can be implemented either using a homography (i.e., known extrinsic parameters [65, 38]) or with a learned projection layer [13]. In contrast, we find it beneficial to remove the intermediate image-level processing and directly predict BEV waypoints, as shown in Fig. 3(b). We replace the upsampling layers with a $3 \times 3$ convolutional layer which fuses the image and speed-based features prior to inputting to three fully-connected final prediction layers. By removing unnecessary processing steps and enabling more expressive image-to-BEV mappings, the proposed planner architecture improves from 2.45 FDE to 2.17 FDE. This improvement comes with a minimal gain in parameters (23.8M vs. 24.1M). Incorporating the geo-conditional module with three adapters further improves trajectory prediction performance to 1.93 FDE with a 24.2M parameter model. We do not find it necessary to leverage more complex, e.g., GRU-based [38], prediction heads.

**Baselines:** While related methods are often studied in simulation [38, 65], we provide several baselines by following publicly available implementations. In particular, we leverage the state-of-the-art monocular agent TCP [38]. To ensure meaningful comparison with single-frame models, we also remove the temporal refinement module. As TCP leverages a control prediction branch in addition to the waypoint prediction branch, we normalize the raw control signals among the different vehicle platforms across the multiple datasets to $[0, 1]$. When comparing with the semi-supervised learning scheme of SelfD [13], we leverage a 10-hour YouTube driving dataset with available city (i.e., region) descriptions. In this manner, we are able to use AnyD to pseudo-label videos that were taken within the 11 cities leveraged in our experiments. Subsequently, we mix the datasets to train a semi-supervised AnyD model, which is then evaluated over our multi-city benchmark.

### 6.1.2 Data Processing and Distributions

To obtain BEV waypoints for training, we standardize formats across three datasets, Argoverse 2 (AV2) [9], nuScenes (nS) [10] and Waymo (Waymo) [7]. The datasets provide post-processed world coordinates for each frame obtained from GPS and other mounted sensors, e.g., LiDAR [10]. We leverage these reported ego-poses to generate a waypoint prediction benchmark. For each frame, we use the global coordinate as an intermediate to get relative positions of the future 2.5s as ground truth waypoints. The conditional command (left, forward, or right) is inferred in a semi-automatic process. First, we extract the preliminary command by thresholding the curvature of the trajectory.

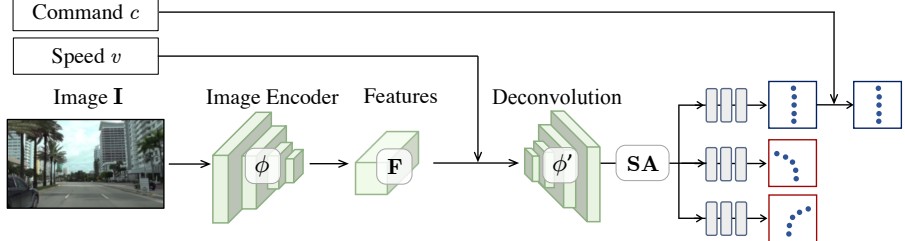

(a) The baseline **BEV Planner** [13] relies on alignment with the image input through upsampling (i.e., to obtain a waypoint heatmap), followed by soft-argmax (SA) [65].

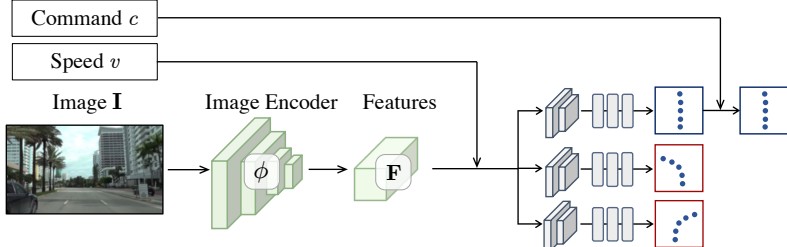

(b) The **proposed planner** model simplifies the architecture such that each command branch directly predicts BEV waypoints without intermediate upsampling, 2D heatmap, or soft-argmax.

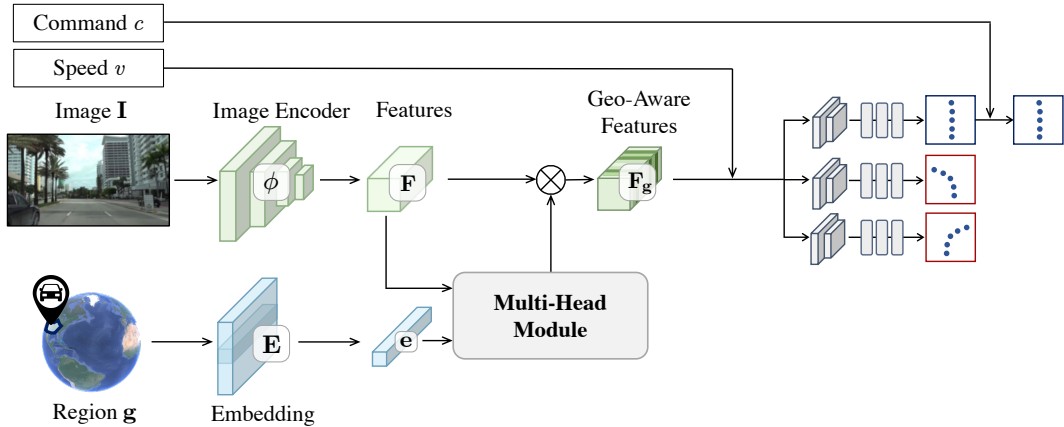

(c) The complete AnyD architecture with geo-aware feature modulation.

Figure 3: **Comparing Network Architectures.** As discussed in the main paper (Sec. 3.4 and Sec. 4.1), our proposed planner architecture abandons the intermediate image-aligned heatmaps and consequent soft-argmax employed by several prior works [65, 13] (top figure) and directly predicts waypoints (middle figure). The proposed architecture improves BEV waypoint prediction results by 12.9% FDE (2.45 vs. 2.17 FDE, shown in Table 1 of the main paper). Incorporating a geo-conditional module (bottom figure) further boosts performance by 4.1% (2.08 FDE) and 11.1% (1.93 FDE) without and with the proposed loss function, respectively.

However, this process cannot detect subtle maneuvers, such as lane changes, which are included in our dataset. Consequently, we manually verify and annotate the initial automatic command predic-

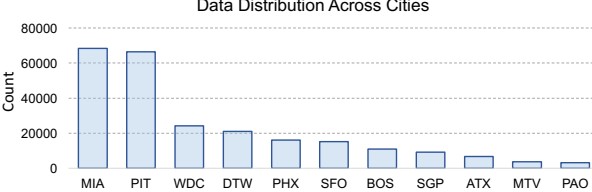

Figure 4: **Imbalanced Training Data Distribution.** Our training data is unevenly distributed across different cities, as often is the case in real-world data (y-axis is sample count).

tions. For our geo-conditional module, we do not require accurate GPS information as we quantize each latitude and longitude into a city-level cluster. Each dataset provides a front-view RGB image and speed (either from the raw CAN bus or from the positioning information), which are inputted into our model as observations.

The data spans 11 cities: Pittsburgh (PIT), Washington, DC (WDC), Miami (MIA), Austin (ATX), Palo Alto (PAO) and Detroit (DTW), Boston (BOS), Singapore (SGP), Phoenix (PHX), San Francisco (SFO) and Mountain View (MTV). The data distribution across different cities is shown in Fig. 4.

### 6.1.3 Evaluation Metrics and Settings

**Open-Loop Evaluation:** Average Displacement Error (ADE) and Final Displacement Error (FDE) are standard metrics for trajectory prediction [75]. We first compute the error within each city, and then average to obtain a balanced average metric. We also generate more fine-grained analysis by providing a breakdown over 11 semantic events in the dataset, as will be further discussed in Sec. 6.2. The extracted events include left turns, forward command, right turns, highway driving, heavy downtown traffic, red traffic lights, stop signs, uncontrolled intersections, pedestrian crossings, rain, and construction zones.

**CARLA Evaluation:** The open-loop evaluation measures the distance of waypoint predictions with real-world human drivers under complex maneuvers, including yielding, merging, and irregular intersections. To further validate our proposed approach, we sought to evaluate AnyD in closed-loop settings where continuous predictions are made in order to navigate a vehicle to a destination along a route. While closed-loop real-world evaluation is challenging due to safety requirements, we leverage the CARLA [82] simulator. Yet, standard CARLA evaluation does not generally involves social and regional behavior that is dynamic across towns. Subsequently, models do not currently incorporate regional modeling, and GPS information is solely used to determine a command at intersections along a route. Hence, motivated by our 11-city real-world benchmark, we introduce a new benchmark where different towns have different traffic behavior. Our benchmark is defined over Town 1, Town 2, and Town 10. We have modified Town 2 for left-hand driving and added pedestrians and vehicles with more aggressive behaviors, e.g., with jaywalking, higher speeds, and closer proximity, to Town 10. We tune the autopilot's controller in order to generate optimal behavior under the novel settings. When performing closed-loop evaluation in CARLA, we also compute a Driving Score (DS) [85], which is a product between the route completion and a penalty based on infractions.

### 6.1.4 Training Protocol

We study the role of our proposed network for scalable deployment use-cases using three training paradigms. To ensure standardized training across both centralized and federated training, we train the model using Stochastic Gradient Descent (SGD) [59]. In **centralized training** (where all the raw observation data is shared in a single server), we use a batch size of 48 and train for 7,500 iterations. We set the initial learning rate to 1e-1, learning rate decay as 0.997, and weight decay as 1e-3. The loss hyper-parameters are set as $\lambda_c = $ 1e-3, $\lambda_g = $ 1e-4, $\lambda_d = $ 1e-4. For **semi-supervised training** settings, model training is done in three stages. We first download a set of online videos

based on their tag which provides city-level information. We then train a supervised model using the same hyper-parameters as in centralized training above, and pseudo-label the unlabeled videos. We train three models from different initial seeds to compute a confidence score (i.e., variance) for each pseudo-label and filter low-confidence predictions. Subsequently, we train our model using the original training dataset combined with the large pseudo-labeled dataset. Here, we set the initial learning rate to 1e-3 and train the model for 500 iterations. Finally, for a fair comparison between the centralized training and the **federated training** settings, we train our federated model for 1,500 synchronous communication rounds. We treat each city as its own 'node' or 'device,' but do not share the private geo-embedding with the server (i.e., we aggregate all model parameters on the server using FedAvg [58] and FedDyn [59] excluding **E**). For each communication round, the model is updated for five local iterations with SGD (in this manner, total iterations remain at 7,500). We further note that we remove the geo-contrastive loss term $\mathcal{L}_{g-ct}$ in the federated learning settings (as this information is not shared among the locations). We keep all other hyper-parameters fixed throughout the training settings.

## 6.2 Additional Ablation and Results

To supplement our findings in the main paper, we discuss four additional results. First, we provide supplementary analysis in terms of FDE (corresponding to Table 2 of the main paper with ADE), context and event-based performance evaluation (Sec. 6.2.1). Second, we perform additional ablation studies regarding the role of the number of heads in the multi-head module, impact of GPS noise over waypoints ground-truth in training, and clusters of the neighborhood-level models (Sec. 6.2.2). Third, we analyze an adaptation experiment to a novel city in Sec. 6.2.3 and show additional qualitative waypoint prediction results in Sec. 6.2.4. Finally, we perform closed-loop evaluation results using the introduced CARLA benchmark.

### 6.2.1 Additional Analysis

**Final Displacement Error:** For completeness, we report FDE results across training paradigms and models in Table 4. While FDE is more challenging as it emphasizes long-term prediction, we observe similar trends among the models and cities compared to the complementary ADE-based analysis in the main paper. Specifically, we demonstrate AnyD to improve over our baseline even using this harsher metric, i.e., from 2.55 to 1.93 average FDE. Semi-Supervised Learning (SSL) provides further gains compared to the centralized AnyD for most cities (excluding ATX and DTW, which show slight under-performance). When compared with Federated Learning (FL), the improvement is less pronounced compared to the gains observed with ADE and FL. While some cities are shown to significantly benefit FL over CL (e.g., BOS, SGP, PHX, SFO, MTV) others do not (e.g., MIA and ATX). ATX is a small dataset with limited diversity and high speed variability. While evaluation becomes less reliable, a low-shot learning setting can also be studied to understand such challenges in the future.

**Event-Driven Analysis:** Table 5 shows a breakdown of driving performance over different events and maneuvers. Here, we see a significant benefit for the introduced regional awareness, higher modeling capacity, and more balanced contrastive objective. For instance, AnyD improves performance over the unevenly distributed commands (a ratio of $2 : 20 : 3$ among left:forward:right in our dataset), for right command from 1.43 ADE with the planner and up to 1.19 ADE with AnyD. Other conditions, such as highway, downtown, crossings, and rain conditions all show improvements as well. These results suggest that the situational adapters are able to accommodate various conditions both within and across geo-locations.

### 6.2.2 Ablation Studies

**Number of Heads :** We investigated the impact of increasing the number of heads in Table 6 on the performance of the model in cities with different amounts of data. We observe that when cities

Table 4: **Analyzing AnyD with Different Training Paradigms (FDE Version).** We analyze the Final Displacement Error (FDE) counterpart of Table 2 in the main paper (which shows Average Displacement Error, ADE). FDE only considers the final waypoint, while ADE considers all waypoints along the predicted route, thus providing complementary analysis. Although both metrics demonstrate similar performance trends, final waypoint prediction is a more challenging task (hence, errors are higher). We analyze the three AnyD training paradigms (CL-Centralized Learning, SSL-Semi-Supervised Learning, and FL-Federated Learning). The results show FDE across the 11 cities in our dataset. Our planner refers to the direct image-to-BEV prediction (without the geolocation information or introduced auxiliary loss terms, see middle architecture in Fig. 3).

| Settings | Methods | Avg | PIT | WDC | MIA | ATX | PAO | DTW | BOS | SGP | PHX | SFO | MTV |
|---|---|---|---|---|---|---|---|---|---|---|---|---|---|
| CL | CIL [65] | 2.55 | 2.20 | 2.71 | 2.85 | 2.47 | 3.05 | 2.02 | 1.69 | 2.14 | 3.03 | 3.07 | 2.86 |
| | CIRL [68] | 2.48 | 2.39 | 2.55 | 2.82 | 2.37 | 3.13 | 2.21 | 1.61 | 2.03 | 2.59 | 2.88 | 2.72 |
| | BEV Planner [13] | 2.45 | 2.30 | 2.08 | 2.66 | 2.64 | 3.25 | 2.04 | 1.75 | 2.09 | 2.51 | 2.87 | 2.73 |
| | TCP [38] | 2.37 | 2.17 | 2.27 | 2.77 | **2.28** | 3.02 | 1.95 | 1.69 | 1.99 | 2.52 | 2.71 | 2.71 |
| | Our Planner | 2.17 | 2.36 | 2.16 | 2.15 | 2.69 | 2.85 | 1.98 | 1.72 | 2.07 | 1.97 | 1.73 | **2.69** |
| | AnyD | **1.93** | **2.19** | **1.97** | **1.94** | 2.48 | **2.83** | **1.80** | **1.55** | **1.96** | **1.88** | **1.62** | 2.80 |
| SSL | SelfD [13] | 1.98 | 2.24 | 2.08 | 2.03 | **2.49** | **2.70** | 1.89 | 1.54 | 1.84 | **1.55** | 1.52 | 2.53 |
| | AnyD | **1.89** | **2.12** | **1.93** | **1.89** | 2.57 | 2.76 | **1.83** | **1.45** | **1.83** | 1.65 | **1.50** | **2.45** |
| FL | FedAvg [58] | 2.63 | 2.65 | 2.85 | 2.69 | 3.68 | 3.43 | 2.55 | 1.76 | 2.31 | 2.87 | 2.67 | 3.13 |
| | FedDyn [59] | 2.14 | 2.48 | 2.34 | 2.44 | 3.42 | **3.10** | 2.35 | 1.53 | 1.86 | 1.86 | 1.85 | 2.55 |
| | AnyD (FedAvg) | 2.23 | 2.51 | 2.21 | 2.12 | 3.52 | 3.40 | 2.04 | 1.58 | 2.09 | 2.43 | 1.97 | 2.79 |
| | AnyD (FedDyn) | **1.92** | **2.25** | **1.91** | **1.94** | **3.26** | 3.29 | **1.90** | **1.37** | **1.68** | **1.31** | **1.49** | **2.28** |

Table 5: **Event-Driven Analysis.** We perform separate evaluations over subsets of our total evaluation benchmark based on semantic events defined over commands and driving conditions. The settings are, in order: left turns, forward command, right turns, highway driving, heavy downtown traffic, red traffic lights, stop signs, uncontrolled intersections, pedestrian crossings, rain, and construction zones. AnyD is shown to benefit from improved robustness across conditions compared to the baseline, i.e., due to the improved modeling capacity.

| Method | Metric | Left | Fwd. | Right | Hwy. | DTown | Red | Stop | Unctrl. | Cross | Rain | Constr. |
|---|---|---|---|---|---|---|---|---|---|---|---|---|
| Our Planner | ADE | 1.60 | 1.01 | 1.43 | 7.66 | 1.12 | 0.66 | **0.63** | 0.77 | 1.36 | 1.37 | 0.76 |
| Full AnyD | | **1.48** | **0.92** | **1.19** | **1.37** | **1.04** | **0.60** | 0.66 | **0.72** | **0.80** | **0.80** | **0.56** |
| Our Planner | FDE | 2.93 | 1.95 | 2.68 | 12.80 | 2.36 | 1.41 | **1.62** | 1.28 | 2.20 | 2.24 | **0.83** |
| Full AnyD | | **2.75** | **1.85** | **2.30** | **2.40** | **2.20** | **1.05** | 1.62 | **1.22** | **1.44** | **1.44** | 1.30 |

Table 6: **Number of Heads in the Multi-Head Module.** We vary the number of heads in the transformer model and compute the resulting model's ADE. We observe a drop in performance beyond three heads (selected throughout the experiments). We demonstrate this to be due to the cities with small (less than 10,000) data samples which may not benefit from the increased modeling capacity.

| # of Heads ($H$) | Large-Data Cities | Small-Data Cities | All Cities |
|---|---|---|---|
| 1 | 1.09 | 1.27 | 1.20 |
| 2 | 1.00 | 1.28 | 1.15 |
| 3 | **0.98** | **1.22** | **1.09** |
| 5 | 1.01 | 1.28 | 1.13 |
| 7 | 1.00 | 1.27 | 1.12 |

have a sufficient amount of data (defined as more than 10,000 frames), the error rate decreases and remains consistently low as the number of heads increases. This can be attributed to the increased model capacity, allowing for more effective feature extraction and improved learning of city-specific patterns. However, when the training data is limited, increasing the number of heads can still result in worse performance on some of the cities, potentially due to overfitting the small data sample. Thus, efficiently increasing model capacity in small-data domains remains a challenge.

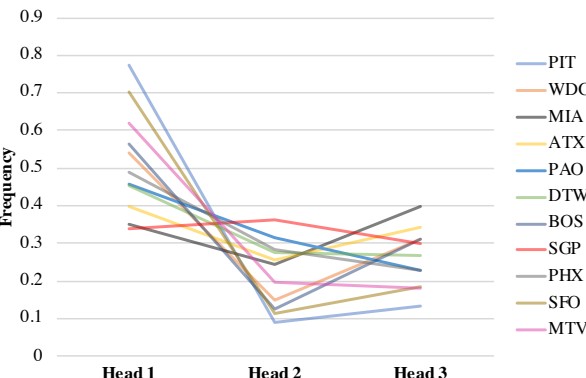

Figure 5: **Head Weight Distribution over Cities**. While the heads and their weights are learned in a data-driven manner, we do find specialization of certain heads across regions. For instance, the unique tropical scenery of Miami (MIA) gives rise to a distinctive pattern among head 1 and head 3, as well as Singapore (SGP).

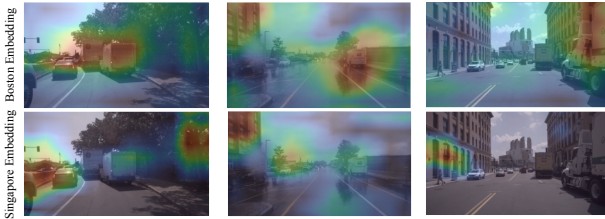

Figure 6: **Attention Visualization in the Geo-Conditional Transformer.** We visualize the attention pattern of the selected head (head with the highest weight) under the same input observations but given different region embeddings. Given Boston embeddings, the weight is shown to concentrate more on the objects in the middle and right portion of the image. While the Singapore embeddings guide attention towards the left portion of the image, which is attributable to the left-hand driving scenario in Singapore.

Fig. 5 studies the frequency of the head of the highest weight. Notably, our results indicate that the distribution in Singapore, where left-hand driving is practiced, differs from that of other cities. Additionally, the unique tropical scenery of Miami gives rise to a distinctive pattern as well.

**Effects of Geo-Conditional Transformer:** Fig. 6 shows the visual effect of the attention pattern of the selected head (head with the highest weight) under the same observations, given different region embeddings. Given Boston embeddings, the head concentrates more on the objects in the middle and right portion of the image. While the Singapore embeddings guide attention towards the left portion of the image, which is attributable to the left-hand driving scenario in Singapore. We note that our supplementary contains additional ablations regarding number of heads in the multi-head attention module and the impact of dataset size on training such higher capacity models.

**Ground-Truth Noise Analysis:** To understand the role of scalable real-world deployment and data collection, we analyze the impact of potential GPS error in Table 7. While our analyzed datasets carefully post-process the reported world coordinates, we envision AnyD deployed across more diverse and potentially noisy settings. We therefore report the performance degradation due to the addition of 1m and 3m Gaussian noise over the training waypoints. Specifically, we find that even with added noise, AnyD obtains decent prediction performance, outperforming prior state-of-the-art models that are trained with clean waypoints, e.g., 2.39 FDE with 3m noise vs. 2.45 FDE for the baseline [65, 68]).

Table 7: **Impact of GPS Noise.** Positioning accuracy (e.g., for obtaining waypoints for training over new locations) is known to be variable. We analyze training with different degrees of Gaussian noise imposed over the ground-truth waypoints ($\sigma$ standard deviation, in meters). We introduce the noise to simulate the data collection in real-world conditions, i.e., in scenarios where the model may be trained over raw GPS measurements without extensive LiDAR-based filtering performed in current autonomous driving datasets. Metrics are averaged over cities.

| Noise | ADE | FDE |
|---|---|---|
| Original | 1.05 | 1.93 |
| $\sigma = 1$ | 1.16 | 2.21 |
| $\sigma = 3$ | 1.28 | 2.39 |

Table 8: **Number of K-Means Clusters vs. Model Performance.** We vary the number of clusters for K-means when obtaining additional finer-grained regions (i.e., neighborhood-level clusters) for each city from publicly available GPS traces [86]. We note that this data may not always be available within all city or country regions. While the additional fine-grained information can further improve our model, we find performance to the plateau beyond three clusters for our data.

| # of Clusters | PIT | WDC | MIA | ATX | PAO | DTW |
|---|---|---|---|---|---|---|
| 1 | **1.15** | 1.27 | 1.60 | 1.13 | 1.65 | 0.96 |
| 3 | 1.16 | 1.09 | 1.11 | **1.10** | 1.42 | **0.95** |
| 10 | 1.21 | **1.06** | **1.09** | 1.22 | **1.39** | 0.98 |

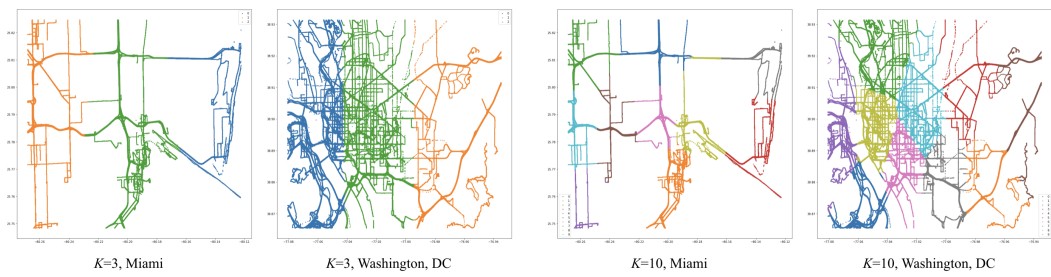

$K$=3, Miami  $K$=3, Washington, DC  $K$=10, Miami  $K$=10, Washington, DC

Figure 7: **Example Neighborhood Clustering by GPS Traces from OpenStreetMap.** We utilize GPS trace data from OpenStreetMap (OSM) [86] to automatically divide cities into sub-regions based on traffic patterns. We leverage K-means clustering to analyze the ability of our model to handle finer-grained regions within cities. Clustering results of Miami and Washington, D.C. with respect to the number of clusters are shown (for 3 and 10 clusters). Despite the coarse clustering, meaningful clusters emerge, e.g., Miami's beach (blue) vs. downtown area (green) in the leftmost $K = 3$ figure.

**Unsupervised Clustering of Regions:** In Table 8, we explore the benefit of finer-grained regional clustering choices, i.e., within each city, when defining **g**. This experiment can uncover potential benefits from modeling intra-city settings. To achieve this, we utilize GPS trace data from OpenStreetMap (OSM)[86] and cluster cities into sub-regions based on traffic patterns. For example, MIA clustering results in semantic regions, e.g., downtown vs. beach areas, with large improvements in prediction performance for the finer-grained model (from 1.60 to 1.11 ADE). Similarly, WDC is clustered into downtown and highway regions, also benefiting performance (from 1.27 to 1.09 ADE). We note that while these examples suggest our model can further benefit model improvement, such clustering data may not be available for many locations, and thus cannot always be assumed.

Table 8 further explores the benefits of finer-grained clustering on AnyD model performance, i.e., within city neighborhoods. To achieve this, we employ GPS trace data from OpenStreetMap (OSM) [86] and divide cities into sub-regions based on traffic patterns via K-means clustering.

Table 9: **Adaptation to a New City.** We fine-tune on additional data from Guangzhou, China [87]. AnyD not only outperforms the baseline model on the new city but also maintains the performance on previously seen cities.

| Cities | Seen Cities (Before → After) | GZ |
|---|---|---|
| BEV Planner [13] | 1.24 → 1.33 | 0.87 |
| AnyD | **1.05 → 1.04** | **0.84** |

Fig. 7 shows example clustering results in MIA and WDC with respect to $K = 3$ and $K = 10$ clusters. For instance, the downtown areas of both two cities can be seen as clusters for both $K = 3$ and $K = 10$. While somewhat coarse in its clustering, we find that this privileged region information (which may not always be available) can introduce additional benefits when used as **g** during model training and evaluation.

### 6.2.3   Adaptation to a New City

In practice, a geo-aware model may be required to learn to drive in a previously unseen region with a significant domain gap. To understand the model performance of AnyD under such learning settings, we extract an additional city (Guangzhou, China) from a different dataset, ApolloScape [87]. In this case, a large domain gap occurs due to the differing social norms and traffic density in China. We mix the new city with the prior 11, and continue fine-tuning the model. Our results in Table 9 indicate that AnyD can learn to drive in the new city while also maintaining similar performance levels for the previously observed cities (i.e., without forgetting). In contrast, the baseline model, which does not incorporate the explicit geo-aware module, has higher error while also impacting performance on prior seen cities.

### 6.2.4   Qualitative Results

We show additional qualitative results in Fig. 8 and Fig. 9 (on the next page). As depicted in the figures, the AnyD generally provides better performance on challenging tasks of navigating across different regions and events, including turns and merging (requires reasoning over traffic directionality) as well as speed limits. Fig. 10 depicts failure cases, which show challenging conditions involving rare rule violations by other vehicles.

### 6.3   Limitations

Despite the multiple publicly available datasets used in our experiments, the diversity in existing benchmarks is still limited, i.e., compared to the vast diversity of geo-locations and events that an agent may encounter in the real-world. Besides Singapore, which provides a challenging generalization use-case , data logs in current datasets are often captured over short drives and are biased towards the US. Thus, our framework requires further validation with larger-scale settings with increased diversity in the future. Here, while our approach for learning a unified model is motivated by human drivers that efficiently learn to adapt generalized skills across locations (including traffic direction), it can be potentially challenging to learn a single model across drastically differing locations. Finally, incorporating various explicit constraints and specifications (e.g., of local traffic rules) could also be studied in the future in order to enable efficient agent adaptation.

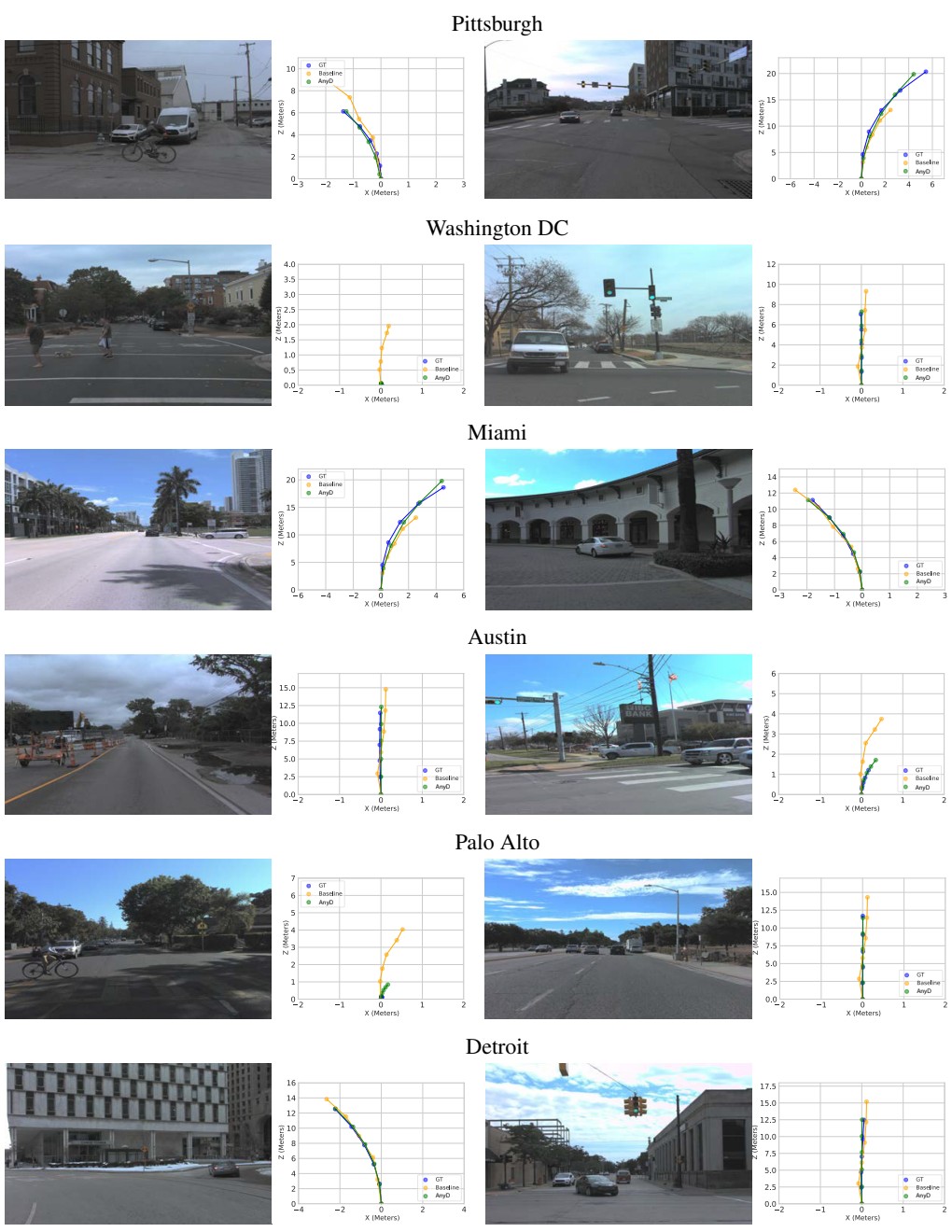

Figure 8: **Qualitative Results across Cities in Argoverse 2 Dataset.** We plot predicted waypoints in the BEV for AnyD, the ground truth trajectory, and the baseline planner model. AnyD is shown to improve reasoning over regional speed and scenarios as well as general navigation and intricate maneuvers.

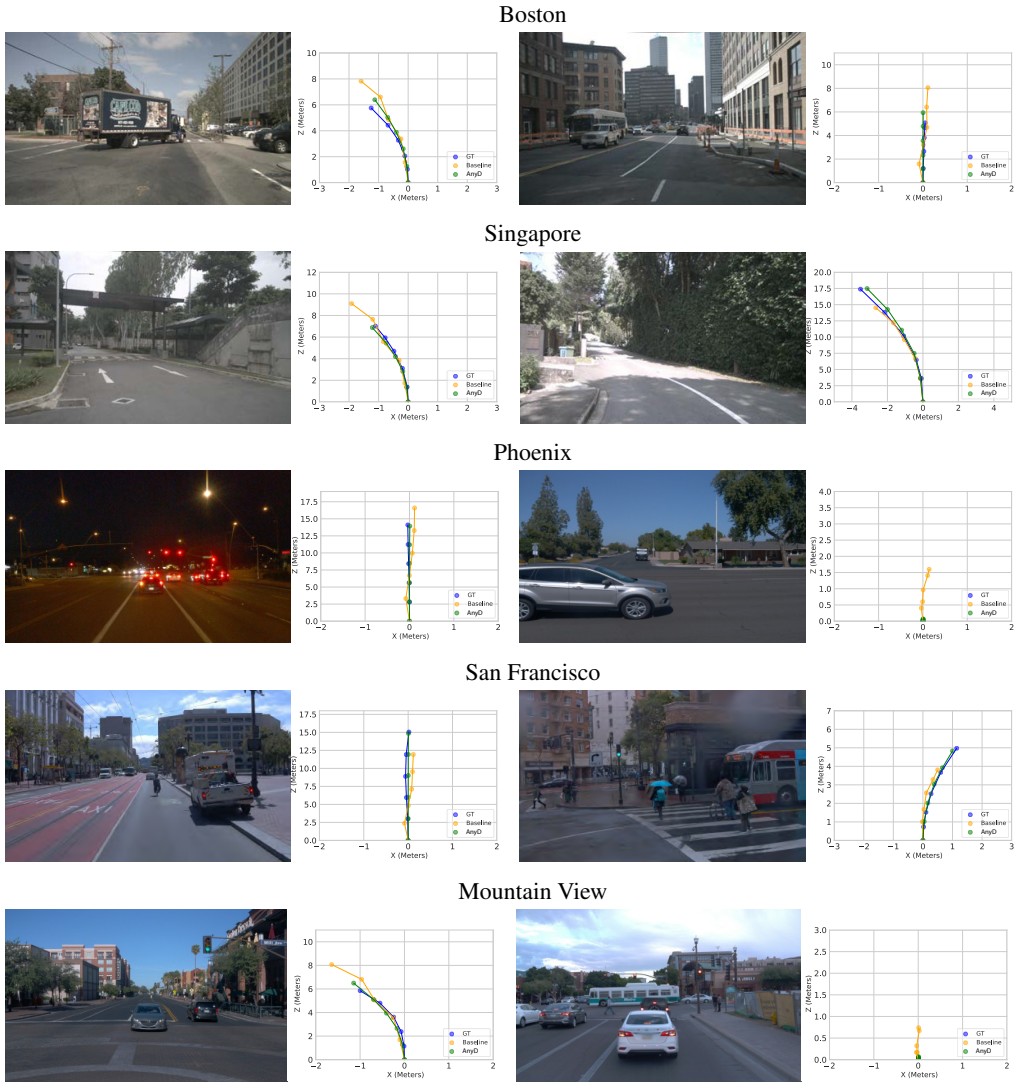

Figure 9: **Qualitative Results across Cities on nS and Waymo Dataset.** We plot predicted way-points in the BEV for AnyD, the ground truth trajectory, and the baseline planner model. AnyD is shown to improve reasoning over regional speed and scenarios as well as general navigation and intricate maneuvers.

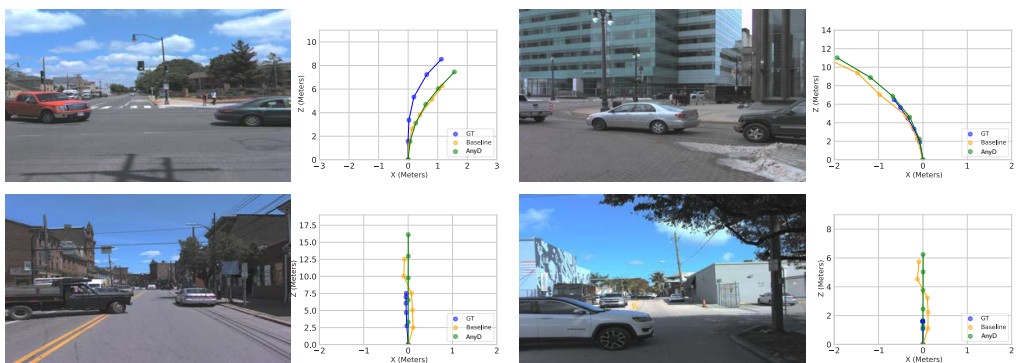

Figure 10: **Example Failure Cases.** Challenging cases where AnyD fails to produce safe driving behavior, often due to dense settings or rare behaviors.

