# OpenReview forum: "Learning to Drive Anywhere"
_robot-learning.org/CoRL/2023/Conference — CoRL 2023 Poster_

### Official Review · Reviewer_tmmM · 2023-07-18

**Confidence:** 4
**Originality:** Good
**Technical Quality:** Good
**Clarity Of Presentation:** Very Good
**Impact:** 4

**Recommendation:**

Weak Accept: I recommend accepting the paper, but will not argue for my recommendation if the majority of other reviewers have a different opinion.

**Review:**

Strengths:
- Paper is well written and clear to understand
- Utilize a combination of public autonomous driving datasets to learn a representation that scales across different deployment scenarios
- Several architectural suggestions for a geo-aware driving agent.

Weaknesses:
- For table 2, some of the baselines compared against that are bolded seem to be close under statistical uncertainty so adding error bars would be quite helpful to quantify how well the proposed method does indeed outperform other baselines.
- Have you tried comparing the visual representation learned with other pre-trained representations? For example, features from a pre-trained Masked Auto-Encoder may perform quite reasonably. Additionally, with the myriad of PeFT approaches (e.g LoRA), these would be compelling in terms of efficiency as well.

Overall, I would recommend a weak accept for this paper.


**Quality Of The Limitations Section:**

Limitations are addressed clearly

**Questions For Rebuttal:**

I would like to hear back regarding the baseline analysis I suggested in the weaknesses section.

Additionally, I had some additional questions for consideration:
1. Several regions around the world are devoid of accurate geographical information. How would your architecture be resilient to noise/inaccurate geographical information.
2. Your paper mentions that the CARLA benchmarks do not generally involve regional modeling. Are there other benchmarks that have this information present within their benchmark? If so, how does your method fare in this setting? In particular, I am concerned if this information provided to the model is privileged and/or unrealistic.

**Robotics Focus:**

Relevant but unlikely to deploy to hardware in near future

**Summary Of Paper:**

This method proposes an approach for conditional imitational learning for driving. In particular, the authors propose an architecture choice of a geo-location based channel attention mechanism and a contrastive objective, which allows them to learn a useful agent for driving in dynamic environments. The method outperforms prior methods in open-loop evaluation in the CARLA simulator.

**Summary Of Recommendation:**

I recommend a weak accept of the paper. I have highlighted some strengths and weaknesses for this work as well as highlighted some questions for consideration. Thank you!

---

### Official Review · Reviewer_pg55 · 2023-07-22

**Confidence:** 4
**Originality:** Good
**Technical Quality:** Good
**Clarity Of Presentation:** Very Good
**Impact:** 3

**Recommendation:**

Weak Accept: I recommend accepting the paper, but will not argue for my recommendation if the majority of other reviewers have a different opinion.

**Review:**

= Contribution

The paper presents an interesting extension of Behavior cloning for autonomous driving. The main challenge in end-2-end autonomy is to learn robust policies. Robustness is difficult to achieve due to many problems involving input data distribution shifts. Here the authors address geographical shift by adding a specialized network branch based on cross-attention to change the policy behavior based on a learned geographical embedding.

= Research Idea

== Quality

The paper is very well executed. The contribution is well-situated and argued. Even though geographical shift is probably not the major issue with end-2-end learning, it tackles an aspect of distributional shift which itself is the main challenge associated with these approaches.

== clarity

The paper is quite clear, though the focus is somewhat not ideally placed. For instance, contrastive learning ends up having no statistical significance in the experiments. This should have been made clear from the start. Otherwise, the reader builds an expectation that it is an essential part of the contribution when it is actually not.

== originality

The paper is rather original in tackling the open-loop question in a very straightforward way. Aggregating data leading to 15 geographical locations leads to interesting results. However I would have expected a focus on: is the data aggregation being leaverage by the agent and how? Is it simply learning better vision features? This could have been studied by freezing the backbone and seeing if the effect of E2E learning are important. Motivating a monolithic approach compared to separated networks.

== significance

The significance of the work I would rate as medium. It definitely shows how to account for regional differences when trained on massive worldwide datasets. However, it could have more impact by treating the questions of closed-loop behavior and the reverse question which is: Is the proposed approach the best way to use a worldwide dataset with E2E learning.

= Strength and Weakness

Strengths
-	Very well-written paper
-	Clarity of argumentation
-	Interesting idea
-	Breadth baseline comparison

Weakness
-	No statistical closed-loop results
-	Does not tackle the main question with sufficient clarity?
o	“ Is this useful?” or could we train on each dataset separately and arbitrate.
-	Too much time spent on contrastive learning with no statistically significant results


**Quality Of The Limitations Section:**

Additional details required

**Questions For Rebuttal:**

- Could you make more clear the benefit of data aggregation in the paper?
- Could you add an experiment where the backbone is frozen?
- Could you add some updated limitation section about open loop result not translating to closed-loop performance?

**Robotics Focus:**

Highly relevant to robotics but no hardware experiments

**Summary Of Paper:**

This paper presents a modification to an end-to-end learning agent that allows it to learn geographically conditioned policies. The claim is that such a feature allows the agent to have an extended operational domain. The key insight underlying the approach is a cross-attentional mechanism that can efficiently switch between the different domains. In the experiments, the authors have trained their agent with three combined publicly available datasets.

**Summary Of Recommendation:**

In summary, the paper would need to drive the point
- Do these modifications improve closed-loop performance?
- What is the benefit of training with an aggregated dataset and why?
(If you can show experimentally that only this kind of regional channel attention allows to learn similarity why excluding differences then the paper would gain much more strength)

---

### Official Review · Reviewer_etZf · 2023-07-26

**Confidence:** 5
**Originality:** Good
**Technical Quality:** Good
**Clarity Of Presentation:** Fair
**Impact:** 3

**Recommendation:**

Weak Accept: I recommend accepting the paper, but will not argue for my recommendation if the majority of other reviewers have a different opinion.

**Review:**

Strength:
1. It is reasonable and interesting to consider the regional factor in the application.
2. The two proposed benchmarks is interesting
3. Experiments show the effectiveness of the proposed method.

Weakness:
1. No statement about open source and thus it is difficult to evaluate how it could really be beneficial for the community.
2. Possiblly due to limited space, the experiment section is unclear and uncomplete. For example:
- In Sec. 4.1 Dataset, the authors mentioned AV2+NS+Waymo and CARLA. However, in the paper,  only results on the former one are reported. All results regarding CARLA are in the supplymental materials.
- In Line298, Youtube videos are mentioned without any context. In Table 2, the setting about SSL is unclear but demonstrated.
- In Table 1 Table 2, and Sec 4.2, it is unclear that the metric is for which dataset. (Though by the minADE, I gusess it is AV2+NS+Waymo).

Thus, I suggest a complete rewriting of the benchmark & experiment section.  The experiments should be divided by setting or datasets in a clearer manner instead of mixed together. For example, if there is no experiments about CARLA in the paper, you should use much less text to mention it.

Missed reference:
There are some SSL works specifically for AD in the community are not mentioned.
[1] Learning to Drive by Watching YouTube videos: Action-Conditioned Contrastive Policy Pretraining. ECCV 22
[2] PPGeo: Policy Pre-training for Autonomous Driving via Self-supervised Geometric Modeling. ICLR 23


**Quality Of The Limitations Section:**

Limitations are addressed clearly

**Questions For Rebuttal:**

1.  My largest concern is about open source. IMHO, the problem and benchmarks proposed in the paper are at least as significant as the model. If the benchmark is not open sourced, I do not think these result tables alone could provide much value to the community.

2. How does your AV2+NS+Waymo and train, val, test split work? How many samples in each set? Is there any overlap among cities? For training, do you just feed all data from diferent source into the model?  There is not enough details about the evaluation protocol.

**Robotics Focus:**

Highly relevant to robotics but no hardware experiments

**Summary Of Paper:**

This paper propose to consider the regional factor of autonomous driving explicitly in the form of extra parameters and network. They build a open-loop real world benchmark and one closed-loop CARLA benchmark. They verify the effectiveness of their methods in both benchmarks.

**Summary Of Recommendation:**

In summary, it is an interesting work with solid experiments, which is why I give a weak accept. However, as mentioned in the weakness part, there are important concerns unsolved. My scores would change according to authors‘ response to these concerns.

---

### Official Review · Reviewer_bBuC · 2023-07-29

**Confidence:** 4
**Originality:** Good
**Technical Quality:** Good
**Clarity Of Presentation:** Good
**Impact:** 3

**Recommendation:**

Weak Reject: I recommend rejecting the paper, but will not argue for my recommendation if the majority of other reviewers have a different opinion.

**Review:**

--

**Quality Of The Limitations Section:**

Limitations are addressed clearly

**Questions For Rebuttal:**

--

**Robotics Focus:**

Relevant but unlikely to deploy to hardware in near future

**Summary Of Paper:**

--

**Summary Of Recommendation:**

--

---

### Official Review · Reviewer_zjyo · 2023-07-30

**Confidence:** 3
**Originality:** Good
**Technical Quality:** Very Good
**Clarity Of Presentation:** Good
**Impact:** 4

**Recommendation:**

Weak Accept: I recommend accepting the paper, but will not argue for my recommendation if the majority of other reviewers have a different opinion.

**Review:**

Praise:
- The paper addresses an important problem in self-driving: the seamless adaptation of a single policy (or a predictor) to the specifics of driving in different geographic regions.
- The idea of using contrastive losses is simple and has strong practical implications.
- The authors perform an extensive evaluation of their method with a decent set of baselines. I find the integration of the method in the federated learning setting particularly appealing.

Concerns:
- Although my general impression of the information flow and the writing style of the work is positive, I believe it could still be improved. I will leave a few specific comments in the “questions” section.
- The main metrics used for comparison do not reflect infractions and traffic rules violations. This is not a strong drawback, of course, since the work focuses on the imitation learning setting solely, and ADE/FDE are the most commonly used metrics in this setting. However, I would encourage the authors to think beyond these simple KPIs, since they often hide driving quality issues.
- It would be interesting to see a more detailed analysis of the limitations, as well as more evaluation of the adaptability of the method. I.e. it is important to point out where practitioners should give up and just keep separate models instead of adapting a single architecture. How and where this boundary should be created, could these regional boundaries be created automatically? To be fair the authors do mention the desire to evaluate their method on a more global scale in the limitations section and the lack of data is, of course, the primary problem here.

**Quality Of The Limitations Section:**

Limitations are addressed clearly

**Questions For Rebuttal:**

- The terms “branched architecture” and “navigation commands” require at least a brief explanation at the spot of introduction (with examples if possible).
- The authors should provide more intuition behind contrastive losses. It is not completely clear why they would help with the imbalanced data specifically.
- Sections on Contrastive Losses require a careful introduction of the notation used (\hat(y)^(c)_i and y_i.), together with how the corresponding inputs are computed: for example, does the framework execute separate forward passes for each command c? These are core contributions, hence a more detailed and easy-to-understand description could be helpful for the reader.


**Robotics Focus:**

Highly relevant to robotics but no hardware experiments

**Summary Of Paper:**

The work presents an approach that allows the adaptation of conditional imitation learning toward the specifics of region-based driving norms. The method comprises two main contributions: i) a cross-attention-based transformer architecture that allows forming queries based on region embedding, and ii) two additional regularizing contrastive losses: one is imposed on navigational commands, and the second is based on the regional head weights. The authors demonstrate the effectiveness of their method on several publicly available datasets, such as nuScenes, Argoverse2, and Waymo, and perform extensive ablation and comparison analysis with a family of existing methods in centralized and federated learning settings.

**Summary Of Recommendation:**

The paper brings a few interesting (albeit not groundbreaking) ideas and is generally pleasant to read.
I believe it has important practical implications, thus I am inclined toward accepting this work.

---

### Author Response · Authors · 2023-08-11
**Response to Reviewers**

We thank the reviewers for their time, effort, and positive feedback. The suggestions helped strengthen the clarity, correctness, and impact of our work. The four reviewers agree that our
work presents an interesting idea in a clear manner with thorough analysis. We performed all of the additional requested ablations and provide an anonymous code repo [link](https://github.com/globaldriver/geco/) (to be released upon acceptance) which can be used to reproduce experiments.

---

### Decision · Program_Chairs · 2023-08-30

**Decision:**

Accept (Poster)

**Comment:**

Scores:  zjyo: Weak Accept,  (bBuC: Weak Reject), etZf:Weak Accept, pg55: Weak Accept, tmmM: Weak Accept

Quality:  The paper focuses on a significant challenge in decision-making in self-driving cars, which involves the seamless adjustment of a single policy to account for the nuances of driving in diverse geographic locations.  This is equivalent to the adaptation to distributional shifts  induced by changes in geographic location. The paper is well executed and the contribution is well-situated and argued.

Clarity: The paper is well written and clear.  The reviewers have acknowledged the efforts made by the authors to improve the manuscript.

Originality: Good. The  paper shows that aggregating data from 11 geographical locations leads to interesting results.

Significance: All reviewers have expressed that the paper contains interesting ideas with solid  experiments.

Pros:
- interesting extension of behavior cloning for geo-aware autonomous driving with solid experiments
- several architectural improvements for a geo-aware driving agent and  breadth baseline comparison
- use of a combination of three public autonomous driving datasets to learn a representation that scales across different deployment scenarios
-  code is available

Cons:
- geographical shifts may  not be the main challenge in autonomous driving E2E learning
- the question  of how data aggregation is  leveraged and whether this it is the best way to make use of a worldwide data set for E2E-learning is not entirely clear